# Feeding Value Assessment of Five Varieties Whole-Plant Cassava in Tropical China

**Mao Li** [1,2,3,†], **Hanlin Zhou** [2,3,†], **Xuejuan Zi** [1,*], **Renlong Lv** [2,3], **Jun Tang** [2,3], **Wenjun Ou** [2] and **Songbi Chen** [1,2,*]

1   Key Laboratory of Ministry of Education for Genetics and Germplasm Innovation of Tropical Special Trees and Ornamental Plants, Key Laboratory of Germplasm Resources of Tropical Special Ornamental Plants of Hainan Province, School of Tropical Agriculture and Forestry, Hainan University, Danzhou 571737, China; limao@catas.cn

2   Tropical Crops Genetic Resources Institute, Chinese Academy of Tropical Agricultural Sciences, Danzhou 571737, China; zhouhanlin@catas.cn (H.Z.); lvrenlong@catas.cn (R.L.); tangjun@catas.cn (J.T.); ouwenjun@catas.cn (W.O.)

3   Zhanjiang Experimental Station, Chinese Academy of Tropical Agricultural Sciences, Zhanjiang 524000, China

*   Correspondence: zixuejuann@hainanu.edu.cn (X.Z.); songbichen@catas.cn (S.C.)

†   These authors contributed equally to this work.

**Abstract:** The feeding value of five varieties of whole-plant cassava (SC5, SC7, SC9, SC14, and SC205) was assessed through analysis of the nutritional composition and in situ ruminal degradability. The results showed abundant nutrients in whole-plant cassava, and the means of starch and crude protein (CP) were 267.7 and 176.8 (g kg$^{-1}$), and ranged from 223.7 g kg$^{-1}$ (SC9) to 296.4 g kg$^{-1}$ (SC14) and from 142.4 g kg$^{-1}$ (SC5) to 195.8 g kg$^{-1}$ (SC9) ($p < 0.05$), respectively. Meanwhile, the moderate neutral detergent fiber (NDF) of whole-plant cassava was also observed and ranged from 266.2 g kg$^{-1}$ in SC9 to 286.6 g kg$^{-1}$ in SC14 ($p < 0.05$). In addition, the trace elements, such as Fe, Mn, Cu, and Zn, in whole-plant cassava were relatively enriched, and their mean concentrations were 135.8, 1225.2, 5.8, and 105.3 mg kg$^{-1}$ ($p < 0.05$), respectively. Both the highest essential amino acid and total amino acid concentrations were obtained in SC7 ($p < 0.01$). The hydrogen cyanide (HCN) content of fresh and dried whole-plant cassava ranged from 76.5 to 131.6 and from 36.0 to 56.7 mg kg$^{-1}$ ($p < 0.05$), respectively. The in situ dry matter ruminal degradability and metabolizable energy (ME) varied significantly ($p < 0.05$) and were consistently lowest and highest in SC9 and SC14, ranging from 50.9% to 80.0% and from 7.5 to 12.3 MJ kg$^{-1}$, respectively. Collectively, all varieties of whole-plant cassava had a high feeding value, as reflected by abundant starch, minerals, amino acid, and water-soluble carbohydrates, while having a low fiber content and HCN toxicity, as well as excellent ruminal digestibility characteristics, and they could be used as a potential feed resource for ruminants.

**Keywords:** whole-plant cassava; amino acids; hydrogen cyanide acid; mineral; ruminal degradability

## 1. Introduction

In the tropics livestock system, C$_4$ grass such as *Pennisetum* is the main roughage for ruminant animals [1]. However, the low nutritional value of these grasses limits the production performance of ruminants. The Hainan Black goat (HNB) is one of the most important ruminants in tropical China, which also faces the issue of a low production performance due to poor nutritional value forage [2]. Therefore, it is urgently necessary to develop high-quality forage resources.

Cassava (*Manihot esculenta* Crantz) is a major food and bio-energy crop grown in tropical areas worldwide [2]. Cassava production in China is rapidly developing due to the increasing demand for its multiple uses, and the plant area is 371,700 ha. Cassava is mainly harvested from roots, to collect more underground portions, and is usually subjected to pruning two or three times each year. The total production of cassava residues is 3,000,000 tons per year in China. The harvested area of cassava is 27,520,043 ha globally,

and it can be predicted that the amount of generated residues will be huge [3]. Fortunately, the use of these cassava by-products for animal feed is one of the alternatives to overcome such problems [4]. Cassava leaves are rich in essential nutrients such as protein, minerals, gross energy (GE), and active components [5–7], and they have been widely used in animal feed (such as goat, sheep, and pig) in many countries [8–13]. In China, research on the utilization of cassava leaves in animals has also made progress, especially in silage making and the effect of goose diets supplemented with cassava leaves [14,15]. In addition, cassava stems contain moderate dietary fibers and starch [16,17]; and cassava tubers are abundant in starch and carbohydrates [18–21]. Cassava leaves, stem, and tubers have been used as feed resources [5,22,23], and the whole-plant cassava represents an ideal feed source, with the potential to aid the sustainable development of local animal husbandry in tropical regions. However, these studies mainly focused on a specific part of the cassava and few variety samples, and the difference in feeding value with whole-plant cassava and among these varieties remains largely unclear. Therefore, to better apply suitable and high-quality cassava feed to animal husbandry, it is necessary to evaluate the feeding value of different varieties of whole-plant cassava.

Cassava is generally used in hay or silage for animal feed and rarely fresh, because it contains hydrogen cyanide (HCN) that causes toxicity, with symptoms of hypoxia and asphyxia [24]. The presence of these compounds is the major factor limiting the utilization of cassava. Therefore, a proper processing procedure is necessary before consumption. Drying is considered one of the best processing methods to reduce the HCN content compared with grating, soaking, and boiling [25,26]. In addition, Oni et al. [27] reported the feeding value of four varieties of cassava in Nigeria, showing that different varieties have significant effects on chemical composition and rumen digestibility, and two varieties showed good potential as protein sources for ruminants. However, to the best of our knowledge, no information is available on the mineral composition, amino acid content, HCN detoxification, and ruminal degradability of different varieties of whole-plant cassava. In the present study, we aimed to evaluate the nutritional value of five varieties of whole-plant cassava in terms of the chemical composition, mineral composition, amino acid, HCN content, and ruminal degradability for goats.

## 2. Materials and Methods

### 2.1. Sample Preparation

A completely randomized design was used in the present study, which was conducted at the Chinese Academy of Tropical Agricultural Sciences (CATAS) in Danzhou, Hainan, China (longitude of 109°30′ E, latitude of 19°30′ N, altitude of 149 m). The average annual rainfall in this area is 1815 mm, with a mean annual temperature of 23.3 °C. Samples were collected from five varieties of whole-plant cassava. The variety accession numbers were as follows: SC5, SC7, SC9, SC14, and SC205.

The varieties were planted during the spring season (15 March) 2020 at CATAS, the planting distance was 80 cm between rows, and 15,000 plants by hectare were in the parcels. Each variety was assigned into five randomly allocated plots, with each plot measuring $5 \times 5$ m. The cassava was fertilized at a rate of 120 kg N, 150 kg P, and 120 kg K ha$^{-1}$, and watering (drip irrigation) was carried out 1 day each week, except when natural precipitation occurred during these days. Weed, pest, and disease control were conducted once a month. The whole-plant cassava was harvested on 15 September 2020, including leaves, stem, and tubers. In addition, 2000 g of whole-plant cassava sample was randomly collected from the five plots of available supply to create five replicates of each variety.

### 2.2. Chemical Analysis

The whole-plant cassava samples were oven-dried at 65 °C for 48 h, then ground and passed through a sieve (1 mm). The weight of the dry matter (DM) was measured after heating the silage samples at 65 °C for 48 h. The oven-dried whole-plant cassava samples were analyzed for their chemical composition. The water-soluble carbohydrates (WSC), crude

protein (CP), neutral detergent fiber (NDF), and acid detergent fiber (ADF) were measured according to the protocols given by the Association of Official Analytical Chemists [28]. The starch content was determined using a total starch determination kit based on the procedures described by Bai et al. [29]. Gross energy (GE) was determined using a bomb calorimeter (Parr6300, Parr Instrument Co., Moline, IL, USA). The HCN content of fresh and dried cassava foliage samples was determined according to the method described by Lei et al. [30]. The minerals in the whole-plant cassava were determined according to the method described by Oni et al. [27] using an atomic absorption spectrophotometer at a wavelength of 550 nm (Buck scientific model 200a; Buck Scientific, East Norwalk, CT, USA). The amino acid content was determined using an amino acid analyzer equipped with a lithium column [31]. All chemical analyses were carried out in quintuplicate.

### 2.3. Relative Feed Value (RFV)

The RFV was developed by the Hay Marketing Task Force of the American Forage and Grassland Council [32]. The RFV of the whole-plant cassava samples was calculated according to this equation, as follows:

$$RFV\ (\%) = \frac{[88.9 \ - \ (0.779 \ \times \ ADF)] \ \times \ (120 \div NDF)}{1.29}$$

### 2.4. Ruminal Degradability Analysis

Five healthy mature Hainan black goats were ruminally cannulated for the comparison of in situ ruminal degradability of DM, CP, NDF, and ADF. Animals were housed on a concrete floor in separate pens and given *ad libitum* access to clean drinking water. The diet of goats included king grass (50%) and concentrate (50%), the composition of the basal diet is listed in Table S1. Goats were adapted for 5 days before the beginning of the experiment. Nylon bags (10 × 5 cm in size, average pore size of 50 μm) were pre-packed with 5 g whole-plant cassava samples using nylon fishing thread. After 72 h of ruminal fermentation, the bags were removed from the rumen, washed with running tap water (15 min) until the rinse was clear, and dried in an oven at 65 °C for 48 h.

After equilibration in the air for 8 h, the bags were weighed, and the residues were transferred to assess the ruminal degradability of DM, CP, NDF, and ADF. The ruminal degradability of DM(DMD), CP(CPD), NDF(NDFD), and ADF(ADFD) was expressed as the ratio of their residues in the ruminal fermentation to their original compositions in the whole-plant cassava. Each experiment was performed in quintuplicate. The whole-plant cassava ruminal degradability of DM, CP, NDF, and ADF was determined as in our previous study [15]. The animal experiment was approved by the Animal Care and Use Committee of CATAS, Hainan Province, China, and performed at the Tropical Animal Research Center of CATAS in August 2017. The experimental procedures were performed according to recommendations proposed by the European Commission (1997) to minimize the suffering of animals and approved by the Animal Care and Use Committee of the Chinese Academy of Tropical Agricultural Sciences (No. CATAS-20140101).

The metabolizable energy (ME) was calculated using the following equation [32]:

$$ME = GE \times DMD \times 0.815$$

where GE is expressed as MJ kg$^{-1}$, and DMD is DM degradability.

### 2.5. Statistical Analysis

The chemical composition and ruminal degradability were analyzed using the general linear models (GLM) of statistical analysis system (SAS) (2013, v. 9.2) in a completely randomized design, to test differences among varieties. The model was as follows:

$$Y_{ij} = \mu + F_i + e_{ij}$$

where $Y_{ij}$ is the observation of different varieties of cassava, $\mu$ is the overall mean, $F_i$ is the fixed effect of the varieties (i = 5), and $e_{ij}$ is the random error. Differences among different varieties of cassava were compared using a probability of difference. The means were compared for statistical significance using Duncan's multiple range tests. A $p < 0.05$ was considered statistically significant.

## 3. Results

### 3.1. Chemical Composition and HCN Content in Whole-Plant Cassava

Table 1 shows the chemical composition and HCN content in all varieties of whole-plant cassava. The chemical composition was affected by the cassava variety. The results showed that the lowest DM was obtained in SC14, and the highest was obtained in SC7, with significant differences ($p < 0.05$). The CP content in SC5 was lower compared with the other varieties, while it was significantly higher in SC9 compared with the other varieties ($p < 0.05$). The mean of ADF content was 136.2 g kg$^{-1}$ and ranged from 130.7 g kg$^{-1}$ (SC5) to 142.8 g kg$^{-1}$ (SC205). The mean of NDF content was 276.9 g kg$^{-1}$ and ranged from 266.2 g kg$^{-1}$ (SC9) to 286.6 g kg$^{-1}$ (SC14). The GE ranged from 17.5 MJ kg$^{-1}$ in SC5 to 18.9 MJ kg$^{-1}$ in SC7, and the mean was 18.3 MJ kg$^{-1}$. The lowest RFV of 254.1 was found in SC7, while the highest was found in SC9, with a value of 274.2 and mean of RFV was 263.2%. Based on the standards assigned by the Hay Market Task Force of the American Forage and Grassland Council, the quality standard of the whole-plant cassava in the present study was graded prime. The WSC and Starch contents had no significant differences, the means were 230.8 and 267.7 201.7 g kg$^{-1}$, and ranged from 201.7 g kg$^{-1}$ (SC14) to 273.5 g kg$^{-1}$ (SC9) and 223.7 g kg$^{-1}$ (SC9) to 296.4 g kg$^{-1}$ (SC14), respectively. The means of HCN found in fresh and dried whole-plant cassava were 97.5 and 44.2 (mg kg$^{-1}$), respectively. The HCN content of fresh whole-plant cassava ranged from 76.5 (mg kg$^{-1}$ FM) in SC9 to 131.6 (mg kg$^{-1}$ FM) in SC7 ($p < 0.05$). Meanwhile, the HCN content after drying ranged from 36.0 (mg kg$^{-1}$ DM) in SC9 to 50.7 (mg kg$^{-1}$ DM) in SC7 ($p < 0.05$).

**Table 1.** Chemical composition of the different varieties of whole-plant cassava.

| | SC5 | SC7 | SC205 | SC9 | SC14 | Mean | SEM | *p* Value |
|---|---|---|---|---|---|---|---|---|
| DM (g kg$^{-1}$ FM) | 283.2 a | 291.5 a | 276.0 a | 281.1 a | 255.8 b | 277.5 | 6.0 | <0.05 |
| CP (g kg$^{-1}$ DM) | 142.4 c | 195.3 a | 173.6 b | 195.8 a | 176.8 b | 176.8 | 9.8 | <0.05 |
| ADF (g kg$^{-1}$ DM) | 130.7 | 137.2 | 142.8 | 134.0 | 136.3 | 136.2 | 2.0 | 0.351 |
| NDF (g kg$^{-1}$ DM) | 269.5 | 285.0 | 277.0 | 266.2 | 286.6 | 276.9 | 4.1 | 0.664 |
| GE (MJ kg$^{-1}$ DM) | 17.5 | 18.9 | 18.7 | 18.3 | 18.3 | 18.3 | 0.2 | 0.219 |
| WSC (g kg$^{-1}$ DM) | 222.3 | 241.8 | 214.7 | 273.5 | 201.7 | 230.8 | 12.5 | 0.533 |
| Starch (g kg$^{-1}$ DM) | 278.9 | 253.1 | 286.3 | 223.7 | 296.4 | 267.7 | 13.2 | 0.246 |
| RFV (%) | 271.7 | 255.3 | 261.2 | 274.2 | 254.1 | 263.2 | 4.2 | 0.437 |
| HCN (mg kg$^{-1}$ FM) | 114.8 b | 131.6 a | 86.4 c | 76.5 c | 78.3 c | 97.5 | 10.9 | <0.05 |
| HCN after drying (mg kg$^{-1}$ DM) | 47.3 a | 50.7 a | 50.3 a | 36.0 b | 36.7 b | 44.2 | 3.26 | <0.05 |

Along the same row, different lowercase letters are significantly different ($p < 0.05$). SEM, standard error of the mean.

### 3.2. Mineral Composition of Whole-Plant Cassava

Table 2 presents the mineral composition of the whole-plant cassava. The mineral composition was also affected by the cassava varieties. The means of Ca, P, K, Mg, Na, Fe, Mn, Cu, and Zn were 9.2 (g kg$^{-1}$), 3.6 (g kg$^{-1}$), 24.9 (g kg$^{-1}$), 7.3 (g kg$^{-1}$), 0.5 (g kg$^{-1}$), 135.8 (mg kg$^{-1}$), 1225.3 (mg kg$^{-1}$), 5.8 (mg kg$^{-1}$), and 105.3 (mg kg$^{-1}$), respectively. The highest contents of Ca and P were found in SC5, while the lowest were detected in SC205 ($p < 0.01$). The content of K ranged from 2.05 g kg$^{-1}$ in SC205 to 3.02 g kg$^{-1}$ in SC9 ($p < 0.05$), while Mg shown the opposite trend, where the highest content was found in SC205 and the lowest was detected in SC9 ($p < 0.05$). The contents of Na and Cu were not significantly different for the varieties of cassava foliage. The lowest Fe content of 93.77 mg kg$^{-1}$ was found in SC5, and the highest content was found in SC205, with a value of 201.95 mg kg$^{-1}$

($p < 0.05$). The highest Mn content was found in SC205, while the lowest was found in SC9 ($p < 0.05$). The highest Zn content was found in SC7 (120.84 mg kg$^{-1}$ DM), whereas the lowest was detected in SC205 (78.40 mg kg$^{-1}$ DM) ($p < 0.05$).

**Table 2.** Mineral composition of the different varieties of whole-plant cassava.

|  | SC5 | SC7 | SC205 | SC9 | SC14 | Mean | SEM | *p* Value |
|---|---|---|---|---|---|---|---|---|
| Ca (g kg$^{-1}$ DM) | 12.2 a | 9.3 b | 7.0 c | 8.4 c | 9.2 b | 9.2 | 0.8 | <0.01 |
| P (g kg$^{-1}$ DM) | 5.3 a | 2.9 b | 3.2 b | 3.1 b | 3.7 b | 3.6 | 0.4 | <0.05 |
| K (g kg$^{-1}$ DM) | 21.9 b | 27.1 a | 20.5 b | 30.2 a | 24.7 b | 24.9 | 1.7 | <0.05 |
| Mg (g kg$^{-1}$ DM) | 7.1 b | 7.8 b | 9.4 a | 5.8 c | 6.6 c | 7.3 | 0.6 | <0.05 |
| Na (g kg$^{-1}$ DM) | 0.5 | 0.6 | 0.4 | 0.4 | 0.4 | 0.5 | 0 | 0.557 |
| Fe (mg kg$^{-1}$ DM) | 93.8 c | 151.2 b | 202.0 a | 96.2 c | 135.8 b | 135.8 | 19.9 | <0.01 |
| Mn (mg kg$^{-1}$ DM) | 1175.5 b | 1150.7 b | 1848.1 a | 933.7 b | 1018.3 b | 1225.3 | 161.6 | <0.01 |
| Cu (mg kg$^{-1}$ DM) | 4.8 | 6.7 | 5.4 | 6.4 | 5.8 | 5.8 | 0.3 | 0.411 |
| Zn (mg kg$^{-1}$ DM) | 109.4 a | 120.8 a | 78.4 b | 112.6 a | 105.31 | 105.3 | 7.2 | <0.05 |

Along the same row, the different lowercase letters are significantly different ($p < 0.05$). SEM, standard error of the mean.

### 3.3. Amino Acid Content of Whole-Plant Cassava

Table 3 shows the amino acid content in all varieties of whole-plant cassava. The means of total amino acids, essential amino acids, and non-essential amino acids were 158.0, 72.9, and 81.4 (g kg$^{-1}$), respectively. The contents of Glu, Gly, Ala, Val, Leu, Phe, Arg, and Pro in the whole-plant cassava were significantly different, and the highest contents were found in SC7. On the contrary, there were no significant differences in the contents of the other amino acids. In addition, the contents of essential amino acid (EAA), non-essential amino acid (NEAA), and total amino acid showed similar trends, where the lowest content was found in SC5 and the highest was found in SC7 ($p < 0.05$). However, there was no significant difference in the ratio of EAA to total amino acid or EAA to NEAA, and their mean ratios were 45.09% and 87.47%, respectively.

**Table 3.** Amino acid composition of the different varieties of whole-plant cassava.

|  | SC5 | SC7 | SC205 | SC9 | SC14 | Mean | SEM | *p* Value |
|---|---|---|---|---|---|---|---|---|
| Essential amino acids (g kg$^{-1}$ DM) | 61.0 c | 82.2 a | 74.2 b | 77.9 a | 61.7 c | 72.9 | 3.6 | <0.01 |
| Thr (g kg$^{-1}$ DM) | 6.8 | 9.2 | 8.2 | 8.5 | 6.9 | 8.1 | 0.4 | 0.231 |
| Val (g kg$^{-1}$ DM) | 8.3 b | 11.2 a | 10.0 a | 10.5 a | 8.3 b | 9.9 | 0.5 | <0.05 |
| Met (g kg$^{-1}$ DM) | 0.7 | 0.97 | 0.71 | 0.94 | 0.7 | 0.8 | 0.1 | 0.267 |
| Iie (g kg$^{-1}$ DM) | 6.8 | 9.1 | 8.2 | 8.7 | 6.8 | 8.1 | 0.4 | 0.182 |
| Leu (g kg$^{-1}$ DM) | 13.2 b | 18.1 a | 16.2 a | 17.1 a | 13.2 b | 15.9 | 0.9 | <0.05 |
| Phe (g kg$^{-1}$ DM) | 8.3 b | 11.0 a | 10.1 a | 10.8 a | 8.3 b | 9.9 | 0.5 | <0.05 |
| Lys (g kg$^{-1}$ DM) | 8.8 | 11.8 | 10.9 | 11.0 | 9.2 | 10.6 | 0.5 | 0.407 |
| Arg (g kg$^{-1}$ DM) | 8.2 b | 10.8 a | 9.9 a | 10.4 a | 8.3 b | 9.7 | 0.4 | <0.05 |
| Non-essential amino acids (g kg$^{-1}$ DM) | 67.8 c | 91.1 a | 84.8 b | 85.0 b | 70.1 c | 81.4 | 10.4 | <0.05 |
| Asp (g kg$^{-1}$ DM) | 14.9 | 19.2 | 18.0 | 17.9 | 16.2 | 17.6 | 0.5 | 0.581 |
| Ser (g kg$^{-1}$ DM) | 6.3 | 8.7 | 7.9 | 8.1 | 7.0 | 7.8 | 0.3 | 0.242 |
| Glu (g kg$^{-1}$ DM) | 18.4 c | 24.2 a | 23.4 a | 21.9 b | 18.5 c | 21.7 | 1.0 | <0.01 |
| Gly (g kg$^{-1}$ DM) | 7.7 b | 10.6 a | 9.5 a | 10.0 a | 7.7 b | 9.3 | 0.5 | <0.05 |
| Ala (g kg$^{-1}$ DM) | 8.8 b | 12.5 a | 11.8 a | 12.1 a | 8.7 b | 1.1 | 0.7 | <0.05 |
| Tyr (g kg$^{-1}$ DM) | 4.7 | 6.4 | 5.6 | 6.2 | 4.7 | 5.6 | 0.3 | 0.39 |
| Pro (g kg$^{-1}$ DM) | 6.9 b | 9.5 a | 8.6 a | 8.8 a | 7.3 b | 8.4 | 0.4 | <0.05 |
| Total amino acids (g kg$^{-1}$) | 131.7 c | 177.0 a | 163.0 b | 167.0 b | 150.0 c | 158.0 | 7.4 | <0.01 |
| Essential amino acids/Total amino acids | 41.2 | 46.4 | 45.5 | 46.7 | 45.7 | 45.1 | 1.0 | 0.533 |
| Total amino acids/Non-essential amino acids | 80.0 | 90.2 | 87.5 | 91.7 | 88.0 | 87.5 | 2.0 | 0.641 |

Along the same row, the different lowercase letters are significantly different ($p < 0.05$). SEM, standard error of the mean.

### 3.4. Ruminal Degradability of Whole-Plant Cassava

Table 4 presents the ruminal degradability of the whole-plant cassava. The means of DMD, NDFD, ADFD, CPD, and ME were 65.3%, 58.3%, 35.1%, 64.8%, and 9.8 MJ kg$^{-1}$, respectively. The highest degradability of DM and NDF of cassava foliage was found in SC14, while the lowest was detected in SC9 ($p < 0.05$). In addition, the ADF ruminal degradability was higher in SC205 and lower in SC5($p < 0.05$). Meanwhile, the CP degradability and ME showed different trends, which ranged from 47.86% (SC9) to 79.70% (SC205) and from 7.53 MJ kg$^{-1}$ (SC9) to 12.30 MJ kg$^{-1}$ (SC14) ($p < 0.01$), respectively.

**Table 4.** Ruminal degradability of the different varieties of whole-plant cassava.

|  | SC5 | SC7 | SC205 | SC9 | SC14 | Mean | SEM | *p* Value |
|---|---|---|---|---|---|---|---|---|
| DMD (%) | 55.8 b | 65.3 ab | 74.7 a | 50.7 c | 80.0 a | 65.3 | 5.5 | <0.05 |
| NDFD (%) | 53.3 b | 58.3 b | 59.1 b | 53.3 b | 67.3 a | 58.3 | 2.6 | <0.05 |
| ADFD (%) | 28.3 c | 35.1 b | 43.3 a | 29.6 c | 39.3 b | 35.1 | 2.8 | <0.01 |
| CPD (%) | 60.0 b | 64.8 b | 79.7 a | 47.9 c | 71.7 a | 64.8 | 5.4 | <0.01 |
| ME (MJ kg$^{-1}$ DM) | 7.9 c | 9.8 b | 11.4 a | 7.5 c | 12.3 a | 9.8 | 2.1 | <0.01 |

Along the same row, the different lowercase letters are significantly different ($p < 0.05$). SEM, standard error of the mean.

## 4. Discussion

Previous studies focused on a specific part of cassava, such as leaves, stems, and tubers. For instance, many researchers have described the chemical composition of cassava leaves, which contain relatively high contents of DM and CP (ranging from 19.1% to 29.2% and from 17.7% to 30.6%), moderate contents of NDF and ADF (ranging from 36.4% to 61.5% and from 25.5% to 44.4%), and various contents of HCN (ranging from 25 to 3689 mg kg$^{-1}$ DM); however, WSC and starch contents were relatively low [11,27,33–36]. In addition, cassava stem has also been used as roughage because it can provide essential fiber components for ruminants, but it lacks minerals, WSC, CP, and GE [16,17]. Moreover, cassava tubers are abundant in starch and carbohydrates and could be used as the main source of energy in animal feed, but their drawbacks are also very obvious, such as lower CP, ADF, and NDF, etc. [18–22]. In the present study, the contents of WSC and starch in whole-plant cassava were higher than in the above reports, while the contents of CP, NDF, ADF, and GE were lower or comparable to previous reports, which may be explained by the whole-plant cassava containing tubers with higher amounts of WSC and starch. Moreover, the proportion of leaves and stems was reduced when the root tubers ripened, so the CP, NDF, and ADF contents decreased. From our results, whole-plant cassava demonstrated a more balanced nutrient profile compared to its leaves, stem, and tubers, containing abundant protein, water-soluble carbohydrates, and starch, while having lower fiber contents [5,17,21,22].

In general, protein and fiber are two of the limiting nutrients in fodder in the tropics, and high-quality fodder contains higher contents of CP and low contents of fiber. Therefore, based on these data, whole-plant cassava can be considered a high-quality fodder. The results showed that the contents of nutrient elements in whole-plant cassava were comparatively high. Overall, the average CP and GE contents in whole-plant cassava were higher than stylo, the most important legume forage used in tropical areas [37]. This finding indicates that these whole-plant cassava can be considered as protein and energy supplements for ruminant basal diets with poor quality. The variation in nutrient contents in whole-plant cassava might be attributed to varietal differences, harvesting period and frequency, fertilization, physiological plant part and fraction, or the sampling method.

Meanwhile, the abovementioned results showed that the whole-plant cassava contained abundant nutrition, while the HCN concentration should be considered. The mean HCN content of fresh whole-plant cassava in the present study was 97.52 mg kg$^{-1}$, which was higher than the Hygienical Standard for Feeds in China (<50 mg kg$^{-1}$) [38]. HCN is the major factor limiting cassava utilization for animal feed. Therefore, proper processing for

HCN degradation is necessary before animal feeding. The drying procedure is considered one of the most effective processing methods for decreasing HCN [26,33]. In the present study, the mean HCN content of dried whole-plant cassava was 44.19 mg kg$^{-1}$, and such a result was similar to previous studies [25,26]. Furthermore, the HCN level of dried whole-plant cassava was close to the standard of the Hygienical Standard for Feeds in China [38], and this ensured the health and safety of the animals.

The mineral composition of cassava leaves has also been previously reported. Oni et al. [27] and Sath et al. [39] reported that the contents of macro-minerals (Ca, P, K, Na, and Mg) in cassava leaves ranged from 4.92 to 13.6, from 1.8 to 2.93, from 1.83 to 11.3, from 0.05 to 0.49, and from 0.65 to 3.5 (mg kg$^{-1}$ DM), respectively, and the contents of micro-minerals (Cu, Fe, Zn, and Mn) ranged from 0.11 to 0.23, from 3.37 to 6.23, from 0.60 to 1.37, and from 0.3 to 2.25 (mg kg$^{-1}$ DM), respectively. Mineral elements are vital for the metabolic processes and health of animals. In the present study, the contents of mineral elements were higher compared with previous reports. Therefore, the values obtained from most of the mineral elements in the five varieties showed that the whole-plant cassava could meet the mineral requirements of ruminants [40].

There have been few studies about amino acid content in cassava. Gomez and Noma [41] reported a low concentration of sulfur-containing amino acid and a relatively high content of arginine in cassava foliage or leaves. Chauynarong et al. [42] and Montagnac et al. [43] reported a similar amino acid composition to our present study and found that the most limiting amino acid in whole-plant cassava was methionine. Nguyen et al. [10] reported a higher amino acid content compared with our present study, and methionine and cysteine were the most limiting amino acids in cassava leaves. However, the amino acid composition and the most limiting amino acid in whole-plant cassava found in the present study were in agreement with earlier reports. The amino acid concentration of whole-plant cassava was more balanced than the egg [44], except for methionine. According to the standard in FAO/WHO [45], EAA accounts for more than 40% of total amino acids, and the ratio of EAA to NEAA is more than 60% for high-quality proteins. Therefore, the proteins in the five varieties of whole-plant cassava in the present study were high-quality proteins. Furthermore, whole-plant cassava is richer in EAA than soybean protein, which is recommended by the FAO [43]. Therefore, from the point of view of amino acid requirement, whole-plant cassava can serve as a source of protein supplementation during soybean deficiency. However, methionine was the most important limiting amino acid in whole-plant cassava, and animal diets need to be supplemented.

Previous studies have reported the ruminal degradability of cassava leaves, and the digestibility of DM, CP, NDF, and ADF ranged from 49.4% to 71.5%, from 45.81% to 65.36%, from 28.1% to 36.6%, and from 19.5 to 27.1%, respectively [27,46–48]. The ruminal degradability of whole-plant cassava in this study was similar to the above-mentioned results. The wide variation in degradability could mostly be attributed to the variable chemical compositions. The results indicated that whole-plant cassava shown good rumen degradation characteristics, which is conducive to the digestion and absorption of nutrients, thereby promoting animal production performance.

## 5. Conclusions

In the present study, we evaluated the feeding value of five varieties of whole-plant cassava. The results demonstrated that the whole-plant cassava contained abundant starch, moderate NDF and ADF, enriched trace elements, and high EAA and total amino acids. The ruminal degradability and ME of whole-plant cassava were also high. In addition, the content of HCN in dried whole-plant cassava was significantly lower compared with fresh samples, and was close to the Hygienical Standard (<50 mg kg$^{-1}$) for feeds in China. Overall, all varieties of whole-plant cassava had a high feeding value, as reflected by abundant starch, minerals, amino acid, water-soluble carbohydrates, and protein, while having low fiber content and HCN toxicity, as well as excellent ruminal digestibility characteristics, and they could be used as potential feed resources for ruminants.

**Supplementary Materials:** The following supporting information can be downloaded at: https: //www.mdpi.com/article/10.3390/fermentation10010045/s1, Table S1: Ingredient and nutrient composition of the concentrates and king grass (g/kg, as feed).

**Author Contributions:** M.L., X.Z., H.Z., R.L., J.T., W.O. and S.C. performed the experimental design work. M.L. and X.Z. conducted the experiments. M.L., X.Z., H.Z., R.L., J.T., W.O. and S.C. analyzed the data. M.L. and X.Z. wrote and revised the manuscript. All authors have read and agreed to the published version of the manuscript.

**Funding:** This study was funded by the Key research and development projects of Hainan province (ZDYF2022XDNY153, HAIKOU2023-050), Agriculture Research System of China (CARS-11) and the Central Public-interest Scientific Institution Basal Research Fund for Chinese Academy of Tropical Agricultural Sciences (No. 1630032022011) and the Ministry of Agriculture and Rural Affairs of the People's Republic of China (16220078).

**Institutional Review Board Statement:** The animal study protocol was approved by the European Commission (1997) to minimize the suffering of animals and approved by the Animal Care and Use Committee of the Chinese Academy of Tropical Agricultural Sciences (No. CATAS-20140101).

**Informed Consent Statement:** Not applicable.

**Data Availability Statement:** The data presented in this study are available upon request from the corresponding author.

**Conflicts of Interest:** The authors declare that they have no conflicts of interest.

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
