# Peer review of "Feeding Value Assessment of Five Varieties Whole-Plant Cassava in Tropical China"

_fermentation, doi:10.3390/fermentation10010045_

Round 1
Reviewer 1 Report
Comments and Suggestions for Authors
Please specify, whether dried or raw whole-plant cassava were used in chemical analysis. In my opinion, the full chemical analysis of dried whole-plant cassava should be taken to account, because only dried plnat could be applied to the animals.
Author Response
Thank you for your valuable comment.We added the details in Chemical analysis and revised the whole paper.

Reviewer 2 Report
Comments and Suggestions for Authors
I recommend that the author thoroughly read the instructions for authors: Author Contributions are missing, for example He should, for example, cite the literature properly, list it in the correct order.
It should, for example, use the units of the SI system (% are not in SI).
I would recommend using units consistently and correctly throughout the document: e.g. in L157 is: 17.45 MJ kg-1, in L166 is: 131.58 (mg kg-1FM) and in L167 is: 36.04 (mg kg-1 DM). (Why not MJ.kg-1). I recommend: 17.5 MJ/kg DM; 131.6 mg/kg FM; 36.0 mg/kg DM.
In addition: For better clarity in the tables, I recommend giving the numbers as abbreviated as follows, or how it is possible to measure them. I would recommend formatting the tables properly. Why is it, for example, in tab. 1 is the P-value presented differently than in the other tables? Letters are usually small and in index.
I recommend in further research to compare the results with cassava silage, the dry matter content of about 28% seems to be high for silage. It would be interesting to compare the reduction of HCN toxicity by drying and ensiling.
Although drying has reduced the content of toxic cyanide, the question is whether this is sufficient. It depends on the time of feeding and the amount of feed. Long-term stress can be life-threatening.
There are 7 self-citations in the manuscript. I mean a lot. Not all are relevant.
​
The whole-plant cassava was harvested on 15 September 2020, but the results are only published now. Why the delay?
Author Response
I recommend that the author thoroughly read the instructions for authors: Author Contributions are missing, for example He should, for example, cite the literature properly, list it in the correct order.
Response: Thank you for your valuable comments. We added Author Contributions at the last of paper.
It should, for example, use the units of the SI system (% are not in SI).
Response: We have revised the Table 1,2,3 and the text in the paper.
I would recommend using units consistently and correctly throughout the document: e.g. in L157 is: 17.45 MJ kg-1, in L166 is: 131.58 (mg kg-1FM) and in L167 is: 36.04 (mg kg-1 DM). (Why not MJ.kg-1). I recommend: 17.5 MJ/kg DM; 131.6 mg/kg FM; 36.0 mg/kg DM.
Response: We have revised the date in the paper.
In addition: For better clarity in the tables, I recommend giving the numbers as abbreviated as follows, or how it is possible to measure them. I would recommend formatting the tables properly. Why is it, for example, in tab. 1 is the P-value presented differently than in the other tables? Letters are usually small and in index.
Response: We have revised the date in tables and the whole paper.
I recommend in further research to compare the results with cassava silage, the dry matter content of about 28% seems to be high for silage. It would be interesting to compare the reduction of HCN toxicity by drying and ensiling.
Response: Thank you for your suggestion. It will be very helpful for our future research.
Although drying has reduced the content of toxic cyanide, the question is whether this is sufficient. It depends on the time of feeding and the amount of feed. Long-term stress can be life-threatening.
Response: We agree with your opinions. The results of this paper only provide reference information on the cyanide content and feeding value of the whole-plant cassava. The amount and feeding method of whole-plant cassava in animal feed still require a large number of feeding experiments to determine.
There are 7 self-citations in the manuscript. I mean a lot. Not all are relevant.
Response: Thanks for your remind, we will reduce them.
The whole-plant cassava was harvested on 15 September 2020, but the results are only published now. Why the delay?
Response: I understand your confusion. Due to the impact of the COVID-19, our research work stagnated and was not handled in time during the review process, so these works were not published in time. But I still believe that our research results have reference value for the diversified utilization of cassava in the future.

Reviewer 3 Report
Comments and Suggestions for Authors
Interesting Manuscript, especially from the practical point of view of using alternative feed. Remarks and comments to the authors in the attached document. Apart from the above, I can only notice that the research is perhaps a little too simple, and I would like to see it include production research.
After acceptance and minor corrections, I can propose the Manuscript for publication in the Journal.

Author Response
Thank you for your valuable comments. We have revised them in the paper. In order to determine the degradation rate, we did not use an in vitro gas production test, but instead used the in vivo nylon bag degradation method. In the future, we will compare the differences in digestion characteristics between two methods for whole-plant cassava.

Round 2
Reviewer 2 Report
Comments and Suggestions for Authors
I am not satisfied with the response. My 3 first reservations about the manuscript, and at the same time the main ones, were not completely accepted.
It follows mainly from the data in the abstract:
L17-18
Cite: “The results found abundant crude protein (CP) ranged from 19.24% in SC5 to 24.58% in SC9 17 (P<0.05), and the moderate neutral detergent fiber (NDF) ranged from 25.70% in SC8 to 32.50% in 18 SC7 (P<0.05).“
Questions:
• Why my recommendations regarding the use of the SI system were not taken into account in the abstract.
• Why different data are used here than in the tables.
• What does the CS8 variant mean if it is not mentioned in the manuscript?
Recommendation:
In terms of statistics, it is recommended to report mean and standard deviation or SEM and no ranged from – to.
Another shortcoming:
References should be listed in order of use, not alphabetically.
Author Response
Reviewer 2 round 2
I am not satisfied with the response. My 3 first reservations about the manuscript, and at the same time the main ones, were not completely accepted.
It follows mainly from the data in the abstract:
L17-18
Cite: “The results found abundant crude protein (CP) ranged from 19.24% in SC5 to 24.58% in SC9 17 (P<0.05), and the moderate neutral detergent fiber (NDF) ranged from 25.70% in SC8 to 32.50% in 18 SC7 (P<0.05).“
Questions:
- Why my recommendations regarding the use of the SI system were not taken into account in the abstract.
- Why different data are used here than in the tables.
- What does the CS8 variant mean if it is not mentioned in the manuscript?
Response: Thank you for your valuable comments. I am so sorry, due to negligence, we did not modify the abstract in the last revision and have now made the necessary changes as required
Recommendation:
In terms of statistics, it is recommended to report mean and standard deviation or SEM and no ranged from – to.
Response: We added description of the means of nutrients in result and abstract.
Another shortcoming:
References should be listed in order of use, not alphabetically.
Response: We have reorganized the references according to the template.